# Quantitative Flow Ratio Is Related to Anatomic Left Main Stem Lesion Parameters as Assessed by Intravascular Imaging

**DOI:** 10.3390/jcm11206024

**Published:** 2022-10-12

**Authors:** Andrea Milzi, Rosalia Dettori, Richard Karl Lubberich, Kathrin Burgmaier, Nikolaus Marx, Sebastian Reith, Mathias Burgmaier

**Affiliations:** 1Department of Internal Medicine I, University Hospital of the RWTH Aachen, Pauwelsstr. 30, 52074 Aachen, Germany; 2Department of Pediatrics, Faculty of Medicine, University of Cologne, University Hospital of Cologne, 50931 Cologne, Germany; 3Faculty of Applied Healthcare Science, Deggendorf Institute of Technology, 94469 Deggendorf, Germany

**Keywords:** coronary artery disease, left main stem, coronary physiology, quantitative flow ratio, intravascular imaging, intravascular ultrasound, optical coherence tomography

## Abstract

Introduction: Previously, an association between anatomic left main stem (LMS) lesion parameters, as described by intravascular ultrasound (IVUS) and fractional flow reserve (FFR), was shown. Quantitative flow ratio (QFR) is a novel, promising technique which can assess functional stenosis relevance based only on angiography. However, as little is known about the relationship between anatomic LMS parameters and QFR, it was thus investigated in this study. Methods: In 53 patients with LMS disease, we tested the association between anatomic assessment using OCT (n = 28) or IVUS (n = 25) on the one hand and functional assessment as determined by QFR on the other hand. LMS-QFR was measured using a dedicated approach, averaging QFR over left anterior descending (LAD) and circumflex (LCX) and manually limiting segment of interest to LMS. Results: The minimal luminal area of the LMS (LMS-MLA) as measured by intravascular imaging showed a consistent correlation with QFR (R = 0.61, *p* < 0.001). QFR could predict a LMS-MLA < 6 mm^2^ with very good diagnostic accuracy (AUC 0.919) and a LMS-MLA < 4.5 mm^2^ with good accuracy (AUC 0.798). Similar results were obtained for other stenosis parameters. Conclusions: QFR might be a valuable tool to assess LMS disease. Further studies focusing on patient outcomes are needed to further validate the effectiveness of this approach.

## 1. Introduction

Left main stem (LMS) disease is present in 5–7% of patients undergoing coronary angiography [1,2]. As the LMS supplies a relevant proportion of myocardium, relevant LMS stenosis is associated with high morbidity and mortality [1,2]. Timely revascularization, which may be represented by coronary-artery bypass grafting (CABG) or percutaneous coronary intervention (PCI) [3,4,5,6,7,8,9,10], is able to reduce this high risk. However, these interventions are associated with intra- and periprocedural risks, which exceed those of procedures performed on other vessels, mainly due to the magnitude of the supplied territory. Therefore, adequate selection of LMS stenoses needing intervention is paramount.

However, assessment of LMS stenoses is often challenging for interventionalists, considering that coronary angiography alone leads to the misclassification of almost one third of patients with LMS stenosis when compared to functional assessment with fractional flow reserve (FFR) [11]. However, an evaluation of the hemodynamic relevance of a LMS stenosis using FFR is technically challenging, due to the need to disengage the guiding catheter and the impossibility of administrating intracoronary adenosine; furthermore, the (frequent) presence of downstream disease may cause false negative results [12,13]. However, a prospective study enrolling 230 patients showed the safety of deferring intervention in patients with FFR > 0.80 [13]. In the light of the technical difficulties in the use of FFR, intravascular imaging, such as intravascular ultrasound (IVUS), has been widely employed in the assessment of LMS disease and received a Class IIa indication in the most recent ESC guidelines on myocardial revascularization [14]. Optical coherence tomography (OCT), as a further intravascular imaging modality, has also been successfully employed in this setting. The cut-offs for defining an ischemia-generating LMS stenosis are derived from studies comparing anatomic stenosis severity, assessed through intravascular imaging (mainly IVUS) on the one hand and the hemodynamic relevance of a lesion as shown by a FFR ≤ 0.80 on the other. Here, studies could show that a minimal luminal area (MLA) ≥ 6 mm^2^ was well associated with hemodynamically irrelevant LMS lesions [15]. In other populations, however, a different MLA threshold of 4.5 mm^2^ has been proposed [16]. Recently, a consensus paper of the European Association of Percutaneous Coronary Interventions (EAPCI) suggested a hybrid approach combining anatomic and physiological assessment. In this document, deferral of LMS revascularization is considered safe when MLA ≥ 6 mm^2^, as this can be considered non-ischemic; revascularization is advocated when MLA < 4.5 mm^2^, whereas in the “grey zone” (MLA 4.5–6 mm^2^) the use of functional assessment such as FFR is suggested [17].

Quantitative flow ratio (QFR) is a novel technique able to assess the hemodynamic relevance of a coronary stenosis based solely on two angiographic projections, without the need for pressure wires or medications [18]. QFR has already shown very promising results in the evaluation of coronary lesions in patients with chronic coronary syndrome [19,20,21] and in the assessment of non-culprit lesions of patients with acute coronary syndromes [22,23,24,25], as well as a good correlation with stenosis geometry [26]. As QFR has been suggested as an angiography-based alternative to FFR, and FFR is associated with stenosis geometry in LMS disease, we aimed to test the association of intravascular imaging with QFR in the context of LMS stenoses, where its performance has not been conclusively assessed yet. This question may be of major interest for interventionalists and may allow to reduce more invasive analyses such as FFR and/or intravascular imaging.

## 2. Materials and Methods

### 2.1. Patient Selection

We retrospectively enrolled 69 patients with LMS disease who had been evaluated by means of intravascular imaging (IVUS or OCT) in the Dept. of Cardiology of the University Hospital of the RWTH Aachen between January 2010 and January 2021. Indication for the use of intravascular imaging was posed by the operator during angiography. The study has been approved by the Ethical Commission of the University Hospital of the RWTH Aachen on 9 May 2022 with protocol nr. 21-204 and is in accordance with the Declaration of Helsinki regarding medical research on human subjects.

### 2.2. Intravascular Imaging: Image Acquisition and Analysis

IVUS and OCT image acquisition have been performed as previously described per standard operating procedures during coronary angiography [27]. Measurement of stenosis parameters such as MLA, minimal lumen diameter (MLD) and percent area stenosis had been performed directly peri-interventionally by the operator by checking and eventually correcting the automatically detected vessel contours. A second intravascular imaging specialist (AM), blinded to the results of previous measurements, performed an independent offline assessment. Averaged values were then used for further analysis.

### 2.3. QFR Analysis

QFR analysis was performed using proprietary software (QAngio suite and QFR, Medis, Leiden, Netherlands) by a certified, experienced user blinded to results of intravascular imaging. Due to the retrospective nature of the study, optimized angiographic projections, as defined in randomized studies [19,20,21], could not be employed; however, in all cases included in the final analysis, image quality was deemed at least sufficient by the operator. Assessment of hemodynamic relevance of LMS was performed after separate reconstruction of both left anterior descending (LAD) and left circumflex (LCx). This aimed to reduce potential sources of error in the determination of LMS-QFR, such as the impact of downstream LAD or LCx disease or any foreshortening or tortuosity in the available projections.

For each artery, two angiographic projections with an angle > 25° were selected from the operator, in order to guarantee optimal visualization of the vessel; the selected projections did not need to be identical for the assessment of LCx and LAD. Then, QFR was obtained, as previously described [19,20,21], by selecting end-diastolic images, drawing the vessel pathway, proofing the automatically detected vessel contour and defining reference diameters as appropriate. For further analysis, contrast-flow QFR was used. For each single-vessel analysis, LMS-QFR was obtained from QFR calculation by manually delimiting the region of interest to the last point immediately proximal to the LAD/LCx bifurcation, using the specific “index QFR” function provided by the commercial software. Any stenosis distal to the LMS bifurcation (i.e., LAD and LCx stenosis) was therefore not included in the measurement of LMS-QFR. These index LMS-QFR values (one obtained from reconstruction of LAD, the other from reconstruction of LCx) were averaged in order to obtain the LMS-QFR value employed in further analysis.

Furthermore, the QFR value of both LAD and LCx as a whole, defined as the QFR value distal to the last stenosis, was registered for each vessel and reported as LAD-QFR or LCx-QFR.

True LMS ostial stenoses have been excluded from the analysis due to the absence of a determinable proximal reference area.

A graphical example of the determination of LMS-QFR is depicted in Figure 1.

### 2.4. Statistical Analysis

Continuous variables were reported as mean ± standard deviation and dichotomic ones as count (percentage). In order to assess the association between stenosis parameters derived from IVUS or OCT on the one hand and LMS-QFR on the other, we performed linear and non-linear regression analysis. The influence of imaging modality on correlation coefficients was assessed with Fisher r-to-z transformation. To analyze the diagnostic efficiency of LMS-QFR in predicting MLA under the accepted thresholds of 4.5 and 6 mm^2^, we performed ROC analysis; diagnostic efficiency was classified as previously described [28]. Optimal cut-offs were defined as the points with the highest Youden index; furthermore, positive and negative predictive values for pre-defined LMS-QFR values (0.75; 0.80; 0.85; 0.90) for prediction of both MLA ≤ 4.5 mm^2^ and MLA ≤ 6 mm^2^ were reported. Comparison of diagnostic efficiency between LMS-QFR and LAD or LCX-QFR, as well as comparison of diagnostic efficiency in different subsets of LMS lesions in predicting MLA under the accepted thresholds, were performed using the DeLong test [29].

All statistical analyses were performed using SPSS v 27.0 (IBM Corp., Armonk, NY, USA). Statistical significance was awarded for *p* < 0.05.

## 3. Results

### 3.1. Feasibility of QFR in the Assessment of LMS Disease

Of the initial 69 LMS lesions with complete angiographic and intravascular assessment, QFR could be performed in 53 (76.8%). Reasons for exclusion were insufficient image quality in 13 cases (18.8%), arrhythmia in 2 cases (2.9%) and others in 1 case (1.4%).

The patient and lesion characteristics of included patients are shown in Table 1.

### 3.2. Association of QFR with Intravascular Imaging

In the 53 LMS lesions in which QFR was feasible, we tested the relationship between LMS-MLA obtained with intravascular imaging (IVUS or OCT) and LMS-QFR. Here, an association could be detected between MLA and LMS-QFR (r = 0.61, *p* < 0.001). Similar associations were detected between other LMS stenosis parameters such as MLD and the percent area stenosis on the one hand and LMS-QFR on the other; these relationships are shown in Figure 2. The association between LMS-MLA and LMS-QFR was not significantly different (*p* = 0.196) in lesions analyzed with IVUS (n = 25; r = 0.47, *p* = 0.017) or with OCT (n = 28; r = 0.66, *p* < 0.001).

After demonstrating an association between intravascular imaging results and QFR in the assessment of LMS disease, we tested the diagnostic efficiency of QFR in predicting anatomically severe LMS disease. Specifically, as different thresholds have been proposed [15,16,17], we tested the diagnostic efficiency of QFR in predicting an MLA ≤ 6 mm^2^ and in predicting an MLA ≤ 4.5 mm^2^. QFR showed, respectively, an excellent efficiency for the prediction of MLA ≤ 6 mm^2^ (AUC 0.919, *p* < 0.001) and a good efficiency for the prediction of MLA ≤ 4.5 mm^2^ (AUC 0.798, *p* < 0.001). ROC curves are reported in Figure 3. Positive and negative predictive values for different QFR values (0.75, 0.80, 0.85, 0.90) of LMS in predicting both MLA ≤ 6 mm^2^ and MLA ≤ 4.5 mm^2^ are presented in Table 2.

LMS-QFR was significantly superior to the averaged QFR values of peripheral LAD and LCx both in the prediction of MLA ≤ 6 mm^2^ (AUCs: 0.919 for LMS-QFR vs. 0.662 for LAD/LCx-QFR, *p* = 0.003) and showed a trend in the prediction of MLA < 4.5 mm^2^ (AUCs: 0.798 for LMS-QFR vs. 0.655 for LAD/LCx-QFR, *p* = 0.085).

### 3.3. Impact of Lesion Localization and Downstream Disease

After showing a very good diagnostic efficiency of QFR in classifying LMS disease, we analyzed whether lesion localization and/or downstream disease may relevantly influence the results of this novel diagnostic tool. Comparisons were again performed using the index LMS-QFR resulting from the average of LAD and LCx reconstructions.

Therefore, we compared diagnostic efficiency of QFR in different anatomic subsets of LMS disease. LMS-QFR showed a similar diagnostic efficiency in assessing the relevance of stenoses in the proximal third of LMS (n = 9, AUC = 1.000 for MLA ≤ 6 mm^2^, AUC = 0.857 for MLA ≤ 4.5 mm^2^) compared to shaft or distal lesions (n = 44, AUC = 0.874 for MLA ≤ 6 mm^2^, AUC = 0.785 for MLA ≤ 4.5 mm^2^, both *p* = ns).

The diagnostic efficiency of LMS-QFR was also not significantly influenced by the presence of downstream disease (defined as a drop ≥ 0.10 in LAD/LCx-QFR compared with LMS-QFR) both in LAD (n = 31 with LAD-disease, AUC 0.889 v.s. 0.888 for MLA ≤ 6 mm^2^ and AUC 0.859 v.s. 0.668 for MLA ≤ 4.5 mm^2^, both *p* = ns) and in LCX (n = 26 with LCX disease, AUC 0.942 v.s. 0.855 for MLA ≤ 6 mm^2^ and AUC 0.783 v.s. 0.788 for MLA ≤ 4.5 mm^2^, both *p* = ns).

## 4. Discussion

The main finding of our study is that LMS stenosis parameters as described by intravascular imaging are related to QFR.

Assessing the hemodynamic relevance of LMS disease is challenging for interventionalists; here, the current gold standard is represented by intravascular imaging (IVUS or OCT), which may be complemented by physiological assessment through FFR in certain cases. Proposed thresholds, however, vary in different studies. Our aim was to assess the feasibility and effectivity of QFR in the context of LMS disease.

### 4.1. QFR Is Feasible in Assessing LMS Disease

First of all, we could show that QFR is a feasible tool to assess LMS disease. In our study, we reported that in 76.8% of exams LMS-QFR was measurable. Therefore, the feasibility of QFR in the assessment of LMS disease may be considered sufficient. This feasibility could be shown in spite of the anatomic challenges posed by LMS disease and, especially, the lack of optimized projections, which have, on the contrary, been employed in previous randomized trials including mainly non-LMS lesions [19,20,21].

In particular, QFR analysis of LMS is especially challenging due to its anatomy. For instance, foreshortening of LMS or overlap and tortuosity of the proximal segments of the epicardial vessels may impede the complete reconstruction of the anatomy needed for QFR calculation. The 18.8% dropouts due to image quality can be considered acceptable provided that the image acquisition was not performed according to a pre-specified protocol. On the contrary, applying dedicated image-acquisition protocols will probably even increase the rate of angiographies on which a QFR assessment of LMS disease is possible by allowing the overcoming of these technical limitations. Another important point is the need to define a proximal reference area, which may be impossible in the presence of low-quality images that do not enable sufficient visualization of healthy proximal segments or very proximal LMS stenosis. The impossibility to determine the proximal reference area is particularly relevant for ostial LMS stenoses, which are therefore unsuitable for QFR analysis and have been excluded from our analysis; here, intravascular imaging and possible invasive physiology remain feasible options.

### 4.2. Anatomic Severity of LMS Disease Shows Association with QFR

More interestingly, LMS-MLA and further stenosis parameters were significantly associated with QFR obtained with intravascular imaging, analogous to similar findings in non-LMS coronary artery disease [26]. This parallels previous findings regarding FFR in the context of LMS disease [15,16] and supports the role of QFR as a possible tool in this subgroup of lesions. Based on previous studies, a LMS-MLA > 6.0 mm^2^ can be considered non-ischemic [30]; this could be detected by QFR with excellent efficiency. Similarly, a LMS stenosis with LMS-MLA ≤ 4.5 mm^2^ is considered to be ischemia-generating [30]; this could be detected by QFR with good efficiency. This remarkable performance of QFR was unaffected by the localization of LMS disease or the presence of downstream disease. This may pave the way to trials assessing QFR as a possible tool to initially stratify LMS disease, in which an initial QFR measurement may allow to rule out a relevant LMS stenosis without the need for more invasive intravascular imaging tools.

Interestingly, the optimal QFR-thresholds derived from our study in order to define a relevant LMS disease differ from the expected values. In fact, a QFR-value of 0.80, which has been consistently shown as the optimal cut-off in studies regarding non-LMS coronary disease [19,20,21], only shows a quite modest performance both in terms of negative and positive predictive values. It has to be remarked that the cut-offs that we defined need to be validated in further, larger studies. However, it is tempting to speculate that this difference between QFR thresholds in LMS compared to non-LMS lesions may be due to the employed method. In fact, normally QFR is measured distal to the end of the stenosis, in the location where normally the FFR wire would be placed. In this study, we limited the region of interest to the sole LMS; therefore, we hypothesize that the part of the drop in flow ratio due to the proximal involvement of LAD/LCX, which is a frequent finding in the context of LMS disease, may be responsible for this difference. Another point that needs to be taken into account is the wide use of OCT in our study cohort; this intravascular imaging modality has been previously employed in the assessment of LMS disease [30], but validated cut-offs for defining the hemodynamic relevance of LMS stenoses are lacking. Moreover, in the light of data regarding non-LMS lesions, where OCT-derived MLA cut-offs for defining hemodynamic significance are approximately 10% lower compared to IVUS, the MLA cut-offs derived from our study need to be deemed as preliminary. Importantly, however, the association between the geometry of LMS stenosis and LMS-QFR did not significantly differ depending on the intravascular imaging modality used.

Furthermore, it has to be noted that a dedicated approach in the assessment of LMS-QFR seems necessary, as the results of such an approach were significantly superior to the sheer assessment of peripheral LAD/LCX QFR values. This may be interpreted in the light of the previously described phenomenon of blood flow shifting in LAD/LCX due to an increased resistance offered by a (proximal) stenosis in the other artery, which may falsely increase the flow ratio [12,13]. Interestingly, in our study the presence of downstream stenosis (defined as a drop ≥ 0.10 in peripheral LAD and/or LCX QFR compared with LMS-QFR) does not significantly affect the diagnostic efficiency of QFR in predicting the hemodynamic relevance of a QFR stenosis.

Nevertheless, in spite of the promising results of this study, in the light of the current data, intravascular imaging and/or invasive physiology remain the gold standards to evaluate LMS disease. In the future, QFR may complement intravascular imaging, possibly in order to rapidly and less invasively screen high-risk LMS disease needing further assessment (and potentially intervention) by means of intravascular imaging. For this, however, larger prospective studies are needed.

### 4.3. Limitations

Although being, to the best of our knowledge, the first study systematically assessing QFR in the context of LMS disease, the relatively small population represents a limitation of our study; therefore, our data need further confirmation in future studies. Moreover, certain lesion subsets have been excluded from analysis due to their unsuitability for QFR analysis, such as lesions presenting large thrombus burden in patients with acute coronary syndromes or ostial lesions, which impede the adequate evaluation of the vessel. Furthermore, due to the relatively small sample size, we are not able to assess the diagnostic accuracy of QFR for all ranges of LMS disease individually; further studies are warranted to clarify this point.

As the decision to treat or defer treatment of LMS disease was based, according to current practice, on intravascular imaging data and not on the (retrospective) QFR analysis, we are unable to conclude on the prognostic value of QFR in this lesion subset. Evaluating the prognostic impact of a QFR-based strategy in LMS disease exceeds the aims of our analysis, the aim of which was to demonstrate the feasibility of QFR in LMS disease and to test its association with anatomic lesion severity. Our data provide valuable insights to plan prospective studies evaluating QFR in LMS disease.

Further research is needed, especially to validate the cut-off values and to compare the prognostic impact of a QFR-based assessment of LMS disease with other decisional tools, such as intravascular imaging and/or FFR. Particularly interesting is the performance of QFR in comparison to FFR in the “grey zone” (LMS-MLA of 4.5–6 mm^2^) suggested by a recent EAPCI position paper [17]. Importantly, in our study an FFR measurement was not performed as part of the assessment of LMS disease, which may represent a further limitation of this analysis.

## 5. Conclusions

Our preliminary data show that anatomic left main stem lesion parameters as assessed by intravascular imaging are related to quantitative flow ratio. However, further research is needed to assess optimal cut-offs and possible pitfalls, as well as the prognostic value of a QFR-based strategy in the management of LMS disease.

## Figures and Tables

**Figure 1 jcm-11-06024-f001:**
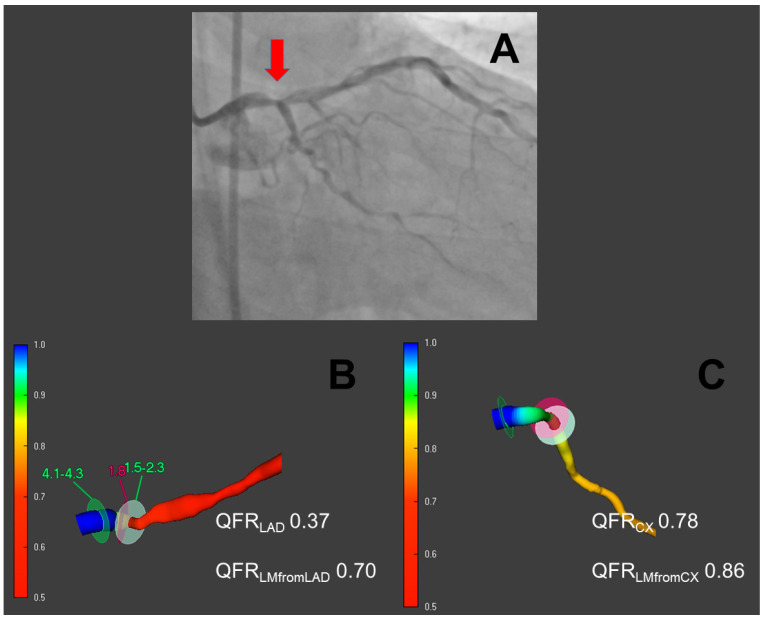
Quantitative flow ratio in the evaluation of LMS disease. In (**A**), an angiographic view of a distal left main stem (LMS) stenosis is depicted (red arrow). In (**B**,**C**), respectively, a three-dimensional reconstruction of the vessel with quantitative flow ratio (QFR) measurement is shown for left anterior descending (LAD) and ramus circumflexus (LCx). LAD and LCX QFR distal to the last stenosis is reported for both vessels. Then, by manually limiting the segment of interest to the LMS only, the LMS-QFR is obtained from each QFR run; these values were averaged and used for further analysis.

**Figure 2 jcm-11-06024-f002:**
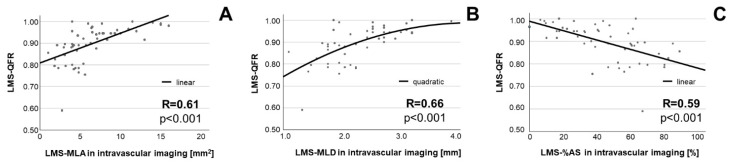
Association between stenosis parameters of LMS and LMS-QFR. Association is shown for MLA (**A**), MLD (**B**) and %AS (**C**). Abbreviations: MLA = minimal lumen area, MLD = minimal lumen diameter, %AS = percent area stenosis.

**Figure 3 jcm-11-06024-f003:**
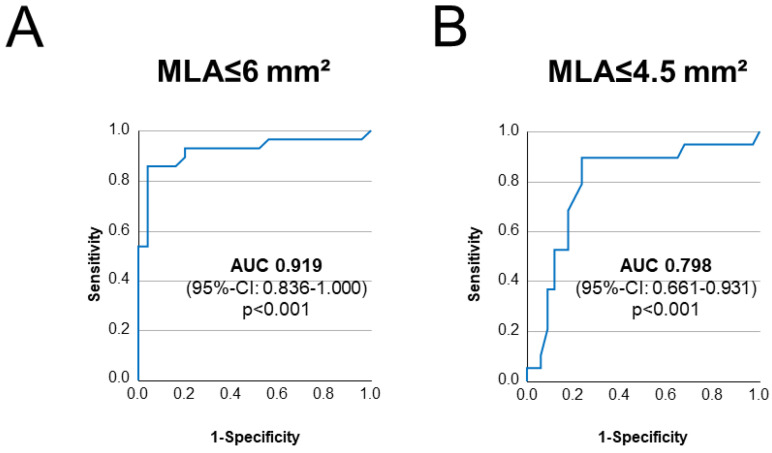
Diagnostic efficiency of LMS-QFR in diagnosing relevant LMS disease. ROC curves depicting diagnostic efficiency of LMS-QFR in predicting anatomic severity of LMS disease, with LMS-MLA ≤ 6 mm^2^ (in (**A**)) and LMS-MLA ≤ 4.5 mm^2^ (in (**B**)), are depicted.

**Table 1 jcm-11-06024-t001:** Patient characteristics at inclusion. Abbreviations: T2DM = type 2 diabetes, MLA = minimal lumen area, QCA-based MLD = minimal lumen diameter based on quantitative coronary angiography, LMS-QFR = quantitative flow ratio of left main stem (see Methods for a detailed description), LAD = left anterior descending, LCx = left circumflex.

	n = 53
Male sex (n,%)	37 (70%)
Age (years)	70.8 ± 9.5
NSTEMI at presentation (n,%)	11 (21%)
T2DM (n,%)	24 (45%)
Hypertension (n,%)	41 (77%)
Nicotine use (n,%)	14 (26%)
Packyears (PY)	10.5 ± 15.0
BMI (g/m^2^)	28.8 ± 4.8
LVEF (%)	47.1 ± 9.5
Lab values	
Cholesterol (mg/dL)	174 ± 56
LDL-c (mg/dL)	109 ± 51
HDL-c (mg/dL)	45 ± 11
Triglycerides (mg/dL)	149 ± 76
HbA1c (%)	6.2 ± 1.1
Lesion characteristics	
Lesion localization	
Proximal	9 (17%)
Shaft	8 (15.1%)
Distal	34 (64.2%)
Diffuse	2 (3.8%)
Medina classification	
1, 0, 0	11 (20.8%)
1, 1, 0	16 (30.2%)
1, 0, 1	11 (20.8%)
1, 1, 1	15 (28.3%)
MLA (mm^2^)	6.5 ± 3.7
MLD in intravascular imaging (mm)	2.43 ± 0.73
QCA-based MLD (mm)	2.22 ± 0.69
Percent diameter stenosis (%)	34 ± 14
Lesion length (mm)	7.9 ± 3.3
Average LMS-QFR	0.90 ± 0.08
LAD involvement (n,%)	33 (62%)
LCx involvement (n,%)	24 (45%)

**Table 2 jcm-11-06024-t002:** Positive and negative predictive values of different LMS-QFR cut-offs in the prediction of anatomic severity of LMS disease.

LMS-QFR	LMS-MLA ≤ 6 mm^2^		LMS-MLA ≤ 4.5 mm^2^	
	PPV	NPV	PPV	NPV
0.75	1.00	0.48	1.00	0.65
0.80	1.00	0.58	0.70	0.72
0.85	1.00	0.62	0.69	0.75
0.90	0.92	0.85	0.65	0.93

Abbreviations: LMS-MLA = minimal lumen area of left main stem, LMS-QFR = quantitative flow ratio of left main stem, PPV = positive predictive value, NPV = negative predictive value.

## Data Availability

The data presented in this study are available on request from the corresponding author. The data are not publicly available due to patient privacy.

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
