# Peer review of "Quantitative Flow Ratio Is Related to Anatomic Left Main Stem Lesion Parameters as Assessed by Intravascular Imaging"

_jcm, 2022, doi:10.3390/jcm11206024_

Round 1

Reviewer 1 Report

In the present manuscript Milzi et al. reported the performance of QFR analysis for the assessment of left main as compared to intracoronary imaging (IVUS, OCT).

The topic is of interest, because angiography-derived FFR is showing its role in the guidance of PCI. The topic is original because LM disease was one of the exclusion criteria in previous studies with QFR.

My comments:

-I suggest to better describe LM disease. In particular describe Medina classification, if the disease involved ostium, mid or distal LM. The major risk may be related to QFR application in cases with diseased LM ostium

-Please stress in the discussion and in the limitations that QFR application in the LM may be biased by several factors and that IVUS/OCT remain the gold standard for LM assessment

-To better describe the correlation between IC imaging and QFR shows also Bland-Altman plot

-Finally, to better describe potential pitfalls in the correlation between QFR and IC imaging please show a graph with the level of agreement (diagnostic accuracy) between the QFR and IC imaging for each range of disease (see Figure 3 in the manuscript PMID: 29449325

Reviewer 2 Report

Ineresting study about a poorly studied, clinically relevant topic

Some minor comments to the authors

-I guess that patients underwent angiography in a stable conditions. Do the authors believe that their findings might be applicable in ACS patients with a high burden of LM thrombi?

-The authors should discuss the potential implications of validating thr prognosis of their QFR methods for assessing LM lesions. SHich sholud be the approach? QFR instead of FFR? instead of imaging techniques? combined?

-What about ostial LM lesions? Any suggestions for the assessment?

Round 2

Reviewer 1 Report

No further comments